# Early Internal Fixation of Concomitant Clavicle Fractures in Severe Thoracic Trauma Prevents Posttraumatic Pneumonia

**DOI:** 10.3390/jcm12154878

**Published:** 2023-07-25

**Authors:** Julia Rehme-Röhrl, Korbinian Sicklinger, Andreas Brand, Julian Fürmetz, Carl Neuerburg, Fabian Stuby, Christian von Rüden

**Affiliations:** 1Department of Trauma Surgery, BG Unfallklinik Murnau, 82418 Murnau, Germany; julia.rehme@bgu-murnau.de (J.R.-R.); korbinian.sicklinger@bgu-murnau.de (K.S.); julian.fuermetz@bgu-murnau.de (J.F.); fabian.stuby@bgu-murnau.de (F.S.); 2Department of Orthopaedics and Trauma Surgery, Musculoskeletal University Center Munich, Ludwig-Maximilians University Munich, 81377 Munich, Germany; carl.neuerburg@med.uni-muenchen.de; 3Institute for Biomechanics, Paracelsus Medical University, 5020 Salzburg, Austria; andreas.brand@bgu-murnau.de; 4Institute for Biomechanics, BG Unfallklinik Murnau, 82418 Murnau, Germany; 5Department of Trauma Surgery, Orthopaedics and Hand Surgery, Weiden Medical Center, 92637 Weiden, Germany

**Keywords:** clavicle fracture, chest wall injury (CWI), thoracic trauma, locking plate fixation, hook plate, disabilities of the arm, shoulder and hand (DASH), Nottingham Clavicle Score

## Abstract

Background: Severe thoracic trauma can lead to pulmonary restriction, loss of lung volume, and difficulty with ventilation. In recent years, there has been increasing evidence of better clinical outcomes following surgical stabilization of clavicle fractures in the setting of this combination of injuries. The aim of this study was to evaluate surgical versus non-surgical treatment of clavicle fractures in severe thoracic trauma in terms of clinical and radiological outcomes in order to make a generalized treatment recommendation based on the results of a large patient cohort. Patients and Methods: This retrospective study included 181 patients (42 women, 139 men) from a European level I trauma centre with a median of 49.3 years in between 2005 and 2021. In 116 cases, the clavicle fracture was stabilized with locking plate or hook plate fixation (group 1), and in 65 cases, it was treated non-surgically (group 2). Long-term functional outcomes at least one year postoperatively using the disabilities of the arm, shoulder and hand (DASH) questionnaire and the Nottingham Clavicle Score (NCS) as well as radiological outcomes were collected in addition to parameters such as hospital days, intensive care days, and complication rates. Results: The Injury Severity Score (ISS) was 17.8 ± 9.8 in group 1 and 19.9 ± 14.4 in group 2 (mean ± SEM; *p* = 0.93), the time in hospital was 21.5 ± 27.2 days in group 1 versus 16 ± 29.3 days in group 2 (*p* = 0.04). Forty-seven patients in group 1 and eleven patients in the group 2 were treated in the ICU. Regarding the duration of ventilation (group 1: 9.1 ± 8.9 days, group 2: 8.1 ± 7.7 days; *p* = 0.64), the functional outcome (DASH group 1: 11 ± 18 points, group 2: 13.7 ± 18. 4 points, *p* = 0.51; NCS group 1: 17.9 ± 8.1 points, group 2: 19.4 ± 10.3 points, *p* = 0.79) and the radiological results, no significant differences were found between the treatment groups. With an overall similar complication rate, pneumonia was found in 2% of patients in group 1 and in 14% of patients in group 2 (*p* = 0.001). Discussion: This study could demonstrate that surgical locking plate fixation of clavicle fractures in combination with CWI significantly reducing the development of posttraumatic pneumonia in a large patient collection and, therefore, can be recommended as standard therapeutic approach for severe thoracic trauma.

## 1. Introduction

Traditionally, displaced fractures of the clavicle have been treated non-operatively [1]. In recent years, a paradigm shift towards an increase in operative treatment occurred [2]. The benefits of surgical clavicle fracture management need to be weighed against the well-known risk of intraoperative or postoperative complications [3]. Although most of the clavicle fractures present as isolated injury, a part of the affected patients also sustains associated thoracic trauma. More than every second patient with an injury severity score (ISS) ≥16 suffers from thoracic trauma [4]. Particularly in polytraumatized patients, unstable chest wall injuries (CWI) are common [5]. Clavicle fractures are common in CWI with an incidence ranging up to 60% [6]. Even higher incidences have been described following open clavicle fractures [7]. Nevertheless, there is a wide range in the severity of the associated CWI. Concurrent rib fractures, for example, may negatively affect the stability of the clavicle fracture [8], since concomitant ipsilateral rib fractures have been reported to significantly increase the extent of displacement of unstable clavicle fractures [9]. While the clavicle is an important stabilizer of the upper quadrant of the chest, displacement and resulting pain can lead to a relevant loss of function of the shoulder girdle and a pronounced deformation of the ipsilateral chest wall [10]. In severe CWI, posttraumatic pneumonia delayed and retained hemothorax or empyema may result in permanent pulmonary restriction, loss of lung volume, and difficulty with ventilation and breathing. This clinical course might affect the treating surgeon’s clavicle fracture management decision [11]. In contrast to the concomitant rib fracture, whose timely surgical stabilization is considered beneficial in preventing posttraumatic pneumonia [12], the effect of early clavicle fracture management in severe thoracic trauma of the seriously injured patient still remains unclear [13,14].

Therefore, the aim of this study was to evaluate clinical and radiological long-term outcomes following operative versus non-operative treatment of concomitant clavicle fractures in severe thoracic trauma to provide a sound general treatment recommendation based on the results of a large patient collection.

## 2. Patients and Methods

In this retrospective study, data from 181 consecutive patients with concomitant clavicle fractures combined with severe thoracic trauma (139 men, 42 women) with a median age of 49.3 (range 16 to 95) years in a European Level I Trauma Center between January 2005 and December 2021 were included. In all cases, patient management was conducted according to the ATLS^®^ guidelines [15]. In group OP, operative management including open reduction and internal locking plate fixation within five days after trauma and in group NO non-operative treatment was performed. The following inclusion criteria were noted: age over 16 years, skeletal maturity, medial, lateral and midshaft fractures according to the Allman classification, combined with three or more unilateral segmental rib fractures or three or more bilateral rib fractures and/or sternal fracture and/or scapula fracture and/or pneumothorax/hemothorax [7,16]. According to Dehghan et al. [17], the exclusion criteria were as follows: Patients with upper airway injury requiring long-term intubation and mechanical ventilation (e.g., tracheal disruption), acute quadriparesis or tetraplegia, head and neck burn injuries, or inhalation burn injuries, dementia or other inability to complete follow-up questionnaires, cases with lack of informed consent from patient or substitute decision maker were excluded from the study. For the item ventilation time, both invasive and noninvasive ventilation were combined. The identical aftercare protocol was conducted with all patients [18]. In the group NO, non-operative treatment included immobilization in a shoulder sling providing patient comfort, especially in the initial phase after trauma [19]. After the symptoms subsided, early physiotherapy was started with passive–assistive exercises including humeral abduction and anteversion to 90° and without weight-bearing for six weeks.

### 2.1. Follow-Up

Follow-up studies were performed at regular intervals, including six weeks, 3, 12, and 24 months, as well as at the most recent visit to the outpatient department. Follow-up assessment included a thorough physical examination, functional evaluation, and diagnostic biplanar conventional radiological studies. The influence of the treatment outcome on patients’ mental and physical health status was assessed using the disabilities of the arm, shoulder and hand (DASH) outcome measure [20] and the modified Nottingham Clavicle Score (NCS; modified version for German patients) [21]. For better understanding, both of the scores were analyzed in a similar way using five choices to answer ranging from “very good” to “very bad”. For better comparability and understanding, the evaluation of the NCS was adapted exactly to the evaluation of the DASH outcome measure: best answer received 1 point and worst answer 5 points. Moreover, Visual Analogue Scale (VAS) was analyzed with a scale ranging from 1 (no pain) to 10 (strongest pain ever experienced). Radiological follow-up was assessed using anterior–posterior and lateral radiographs. According to Fisher et al., osseous healing was defined as formation of bridging callus at all four cortices, and absence of fracture lines [22]. Outcome measures were presented as mean and standard error of the mean (SEM).

### 2.2. Statistical Analysis

Statistical comparisons between groups were conducted using IBM SPSS^®^ Statistics for Windows (IBM Corp., Armonk, NY, USA). Based on ordinal data scales, the Mann–Whitney U test was applied for comparisons of clinical scores (modified NCS, ISS, DASH) and for pain assessment (VAS). Additional clinical measures were compared using either the Mann–Whitney U test (time on ICU, ventilation time, time in hospital) or chi-squared test (Allman classification, polytrauma, pneumonia rate, gender, injury side). The two-sample t-test was used to compare age differences between groups. The level of significance was set at *p* < 0.05.

## 3. Results

An overview on patients‘ general data is displayed in Table 1. In 116 out of 181 patients, clavicle fractures were treated operatively using locking plate fixation (Figure 1a–c) with or without hook. In the remaining 65 cases, non-operative therapy was used. The mean ISS was 17.8 ± 9.8 in group 1 and 19.9 ± 14.4 in group NO (*p* = 0.93).

One-hundred-and-four patients suffered from the clavicle fracture as part of polytraumatization. High-energy trauma was the cause of the thoracic trauma in 132 patients. Twenty-one patients had a car accident, thirty-eight patients had a motor bike accident, fifty-two patients had a bicycle accident, eight patients had a ski accident, five patients suffered a fall from horseback, and the remaining eight patients suffered a fall from a height over 3 m. Simple falls occurred in twenty-three patients and a blunt impact in five patients. Twenty-one patients could not be categorized clearly. According to the Allman classification, 123 fractures were located in the mid third, 54 fractures in the lateral third, and 4 fractures in the medial third. A significant difference was found between both groups regarding the midshaft location. In four patients, open clavicle fractures occurred. In about 60% of patients in both treatment groups, the left side of the thorax was affected.

There was no significant different concerning the range of time on ICU between the treatment groups, but operatively treated patients demonstrated significantly less ventilation time (*p* = 0.04) and significantly more days in hospital (*p* = 0.04) compared with non-operatively treated patients. Furthermore, the rate of posttraumatic pneumonia was significantly higher following non-operative treatment (*p* < 0.001) (Table 2). Since noninvasive ventilation was performed not only on ICU but also on the normal ward, a longer ventilation time was observed than time on ICU in the non-operative group. 

Clinical long-term results using the DASH, modified NCS, and VAS are presented in Table 3. Median follow-up was 7 years (1–16 years) after trauma. Due to a loss of follow-up, data from 78 out of 181 patients (43%) could be included in the study (group OP: 56 patients; group NO: 22 patients). To ensure that potential posttraumatic pain was due to the trauma and not to other factors unrelated to the accident, the VAS was evaluated 12 months after the accident.

Due to the above loss of a follow-up description, conventional radiographs were analyzed in 48 patients of group OP, in 6 patients of group NO after one year (Figure 1c), in 56 patients of group OP, respectively, and in 14 patients of group NO available for a final follow-up two years after trauma. Radiological results did not demonstrate any significant difference between the treatment groups (*p* = 0.28). Aseptic clavicle nonunion was observed in three patients of each treatment group one year after surgery and in one patient of each treatment group two years after surgery.

## 4. Discussion

The aim of this study was to evaluate if there is a significant difference between operative and non-operative treatment of clavicle fractures in combination with severe thoracic trauma. the lung function is especially known to have a decisive impact on the overall clinical outcome [17]. An isolated unstable clavicle fracture is known to be causative for an ineffective respiration and oxygenation [23,24,25]. Therefore, in combination with severe thoracic trauma, it may lead to increasing organ dysfunction and pulmonary failure [26]. Inadequate treatment of severe thoracic trauma might increase the risk of developing life-threatening complications, long-term morbidity, and elevated mortality rates [27,28,29]. In summary, the entire thoracic trauma in combination with the clavicle fracture represents the challenge for the treating surgeon. Operative treatment of CWI is known to be associated with advantages regarding the clinical course and long-term outcome following polytrauma [30,31]. The positive effects of rib fixation in preventing pneumonia following CWI have been well established [32]. In contrast, no studies have yet been performed on this topic for concomitant clavicle fractures, although surgical fixation of the clavicle fracture appears to be much easier and less likely to cause complications compared with osteosynthesis of rib fractures. To our knowledge, there was no information on the long-term effect following operative versus non-operative treatment of concomitant clavicle fractures associated with CWI in the literature yet. So far, the most interesting finding of the current study for daily clinical practice was the significantly lower pneumonia rate in the operative group compared to the non-operative group.

Nevertheless, no significant differences between the treatment groups concerning the remaining clinical and radiologic long-term results could be evaluated. Previous literature highlighted nonunion rates of up to 15% following non-surgical clavicle fracture management [33,34,35]. Interestingly, the current patient cohort demonstrated a nonunion rate of only 1% in the operative group versus 7% in the non-operative group.

Another important finding of the current study was the effect on clinically relevant outcome parameters during the initial stay in hospital and in the long-term clinical course. The ICU stay was comparable, but the ventilation time was prolonged compared to recently published data from polytrauma patients without thoracic trauma [36]. This was in line with a meta analysis provided by Leinicke et al., reporting that patients demonstrated shorter ventilation time following early internal clavicle fracture fixation than following non-operative treatment [30]. However, in the current study, the duration of treatment in hospital was significantly higher in the operative group. This was unexpected and in contrast to Leinicke et al. reporting that both the duration of treatment in the ICU and the total length of stay in the hospital were significantly decreased by operative treatment. One reason for this may be that these parameters might have been influenced by other injuries except the thoracic trauma or by certain comorbidities. On the other hand, in the current study, operative clavicle fracture treatment was only performed when the patients’ overall clinical status was good enough for surgery. Furthermore, it was accepted as a relatively minor operative procedure compared to polytraumatization [37].

Generally, data of the current patient cohort corresponded with other studies reporting the epidemiology of concomitant clavicle fractures combined with severe thoracic trauma [17,23,38]. Mainly traffic accidents with high-energy trauma could be identified as cause of severe thoracic trauma [36,39,40].

According to the Allman classification, the study groups were statistically different regarding the midshaft location. Additionally, all open fractures were treated operatively and confirmed the current golden standard in open clavicle fracture management [41]. Furthermore, several authors recommended internal stabilization of clavicle fractures in thoracic trauma, particularly in the case of displaced fractures and associated serial rib fractures or flail chest injury [10]. Another subject was the heterogeneous patient collection regarding the decision for operative versus non-operative treatment. However, even upon reviewing the individual cases, the groups were comparable with respect to the ISS.

According to the results of this study, we recommend to include timely internal locked plating of concomitant clavicle fractures in the standard treatment regime after severe thoracic trauma aiming to prevent the development of pneumonia in the context of a severe posttraumatic course after severe CWI. 

### Study Limitations

The current study had limitations, such as its retrospective nature. First of all, the follow-up period of seven years after surgery was pleasingly relatively long and comparable with Nowak et al. [42], but unfortunately, it was also associated with a relatively high loss of follow-up. Furthermore, the number of patients treated non-operatively in the long-term follow-up was significantly lower than the number treated operatively, which complicated the comparability of the results. Accordingly, it was not possible to randomize age, gender, and indication for the treatment concepts. The advantages of the study were the exceedingly large cohort size and the fact that all patients were treated by the same team of surgeons in the same hospital according to the same treatment and aftercare protocol. Considering that the cases of clavicle fracture associated with CWI are relatively rare and difficult to collect and that only few cases are available in the literature, the results of this study with a long-term follow-up of consecutive patients may be highly relevant.

## 5. Conclusions

The clinical and radiologic long-term results of this study could demonstrate that timely open reduction and internal locking plate fixation of concomitant clavicle fractures associated with CWI significantly decreased the development of posttraumatic pneumonia in a large patient collection and, therefore, may be recommended as a standard surgical approach in cases of severe thoracic trauma with concurrent clavicle fracture.

## Figures and Tables

**Figure 1 jcm-12-04878-f001:**
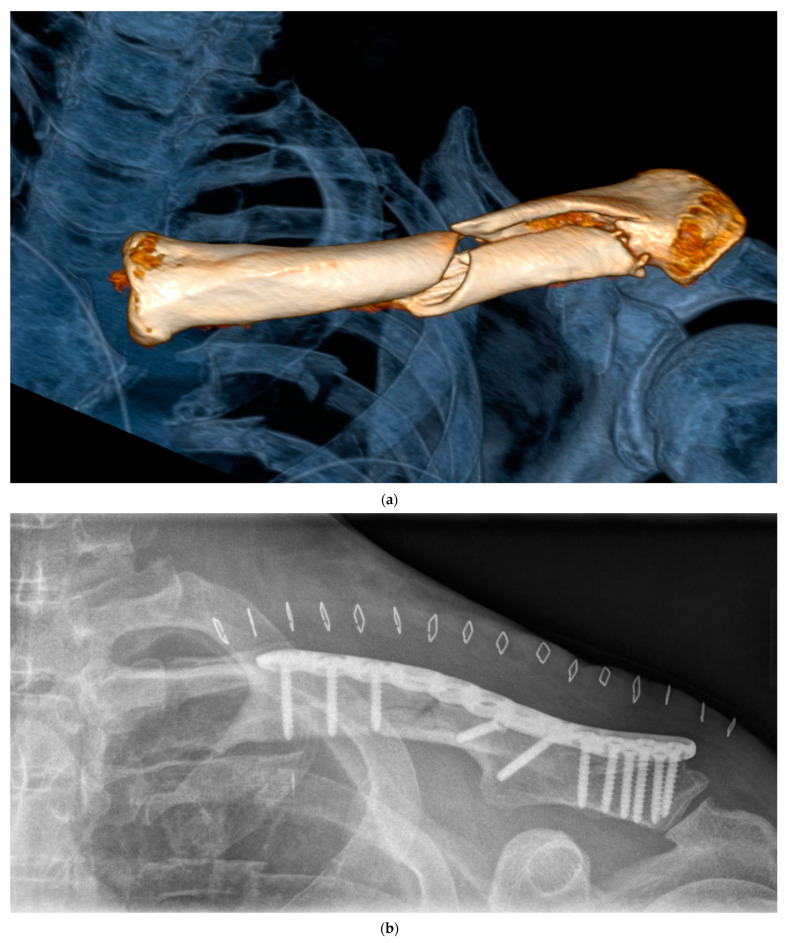
(**a**) Three-dimensional reconstructions of the polytrauma whole-body-computed tomography scan in a 51-year-old male patient after bicycle accident: Severe thoracic trauma including serial rib fractures and concomitant ipsilateral displaced multifragmentary clavicle fracture. (**b**) Postoperative anterior–posterior radiograph after internal precontoured locking plate fixation of the clavicle via longitudinal approach (skin clips). (**c**) Programmed radiological control demonstrated osseous healing one year after trauma. The serial rib fracture also healed after non-operative treatment.

**Table 1 jcm-12-04878-t001:** Overview on patients‘ general data.

	Operative (Group OP)	Non-Operative (Group NO)	*p*-Value
Patients [number]	116	65	
Gender [male/female]	97/19	42/23	0.004
Age [years]	48 ± 14	53 ± 20	0.06
Injured side [right/left]	47/69	24/41	0.63
Allman classification [midshaft/lateral/medial]	88/25/3	35/29/1	<0.01
ISS (mean ± SD) [points]	17.8 ± 9.8	19.9 ± 14.4	0.93
Polytrauma [yes/no]	70/46	34/31	0.29

**Table 2 jcm-12-04878-t002:** Parameters related to the hospital stay.

	Operative (Group OP)	Non-Operative (Group NO)	*p*-Value
Time on ICU [days]	9.1 ± 8.9	8.1 ± 7.7	0.25
Ventilation time [days]	7.5 ± 10	13.6 ± 9	0.04
Time in hospital [days]	21.5 ± 27.2	16 ± 29.3	0.04
Pneumonia [yes/no]	2/114	9/56	<0.001

**Table 3 jcm-12-04878-t003:** Clinical long-term results.

	Operative (Group OP)	Non-Operative (Group NO)	*p*-Value
Follow-up period [years]	7 (1–15)	8 (1–16)	
DASH [points]	10 ± 17	13.7 ± 18.4	0.39
Modified NCS [points]	17.3 ± 7.5	19.4 ± 10.3	0.63
VAS [points]	1.9 ± 2.5	2.4 ± 2.3	0.29

## Data Availability

The datasets analyzed during the current work are available from the corresponding author upon reasonable request.

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
