# Peer review of "Early Internal Fixation of Concomitant Clavicle Fractures in Severe Thoracic Trauma Prevents Posttraumatic Pneumonia"

_jcm, 2023, doi:10.3390/jcm12154878_

Round 1

Reviewer 1 Report

Dear Authors,

I had the honor to review your work but detected some major areas that mandate a significant make-over of the -otherwise interesting- manuscript. Most importantly:

a) Abstract: cohort =  188 patients (43female, 145 male)

Manuscript: cohort = 181 patients (42 female, 139 male)

b) Abstract: operated patients 122 Manuscript: operated patients 116.

c) Scoring systems NCS (mainly) and DASH became popular later in time than the prospective data collection of the author's institution at 2005. In which way were the patients evaluated then ? Did all patients underwent dual evaluation with the above 2 scoring systems ? NCS system is generally used after 2010. Please explain more.

d) Pneumonia is a multi-aspect diagnostic entity invloving an infectious parameter. Please elaborate more on the actual diagnosis of type of pneumonia (ie positive cultures on BALF or peripheral blood) antibiotic coverage of operated patients vs non-operated.

c) Severe thoracic trauma in both penetrating and blunt types includes parenchymal injury which affects outcomes and parameters eg. length of ICU & mech.ventillation duration and morbidity. What is the percentage that received intervention for these intrathoracic injuries ? What about concominant non-thoracic acute procedures, for example exploratory laparotomy, maxillofacial, ortho- or neurosurgical operations. These are important aspects to consider since outcomes depend on them. Please clarify.

The text is well presented and with proper use of the english language.

Author Response

Dear Authors,

I had the honor to review your work but detected some major areas that mandate a significant make-over of the -otherwise interesting- manuscript. Most importantly:

  1. a) Abstract: cohort = 188 patients (43female, 145 male)

Manuscript: cohort = 181 patients (42 female, 139 male)

  1. b) Abstract: operated patients 122 Manuscript: operated patients 116.

Answer: Dear reviewer, we would like to apologize that an earlier intermediate version of the abstract accidentally ended up in the manuscript. After careful multiple analysis of the data sets for completeness, the data mentioned in the manuscript are the correct ones. We have replaced the abstract with the latest version accordingly. Thus, the data from Abstract match the correct data in the manuscript.

  1. c) Scoring systems NCS (mainly) and DASH became popular later in time than the prospective data collection of the author's institution at 2005. In which way were the patients evaluated then ? Did all patients underwent dual evaluation with the above 2 scoring systems ? NCS system is generally used after 2010. Please explain more.

Answer: Dear reviewer, thank you very much for this valuable question. Because these two scores were widely available and accepted at the time we began the retrospective study, we applied them retrospectively to all cases. We removed the term "prospectively collected" in the M+M section accordingly.

  1. d) Pneumonia is a multi-aspect diagnostic entity invloving an infectious parameter. Please elaborate more on the actual diagnosis of type of pneumonia (ie positive cultures on BALF or peripheral blood) antibiotic coverage of operated patients vs non-operated.

Answer: Dear reviewer, thank you for the important comment. The presence of pneumonia was defined in all cases by the presence of positive peripheral blood cultures and corresponding abnormalities on chest X-ray. In all cases, a resistance-appropriate antimicrobial therapy regimen was maintained, regardless of whether the clavicle fracture was surgically stabilized or not.

  1. c) Severe thoracic trauma in both penetrating and blunt types includes parenchymal injury which affects outcomes and parameters eg. length of ICU & mech.ventillation duration and morbidity. What is the percentage that received intervention for these intrathoracic injuries ? What about concominant non-thoracic acute procedures, for example exploratory laparotomy, maxillofacial, ortho- or neurosurgical operations. These are important aspects to consider since outcomes depend on them. Please clarify.

Answer: Dear reviewer, thank you for the question. The periods of mechanical ventilation and ICU stay in the group comparison are presented in Table 2. No other acute non-thoracic procedures such as exploratory laparotomy or orofacial surgery were performed in the studied patient population.

Reviewer 2 Report

Thank you for the opportunity to review your work. I have only minor comments:

- Title: Please revise to imply the main findings of your study. A statement title is more acceptable than questionable one.

- Results: Figures of treatment might not be necessary. 

- Discussion: What are the implication of your findings in more general surgeon, since the readers might not be orthopaedic surgeon.

Author Response

Thank you for the opportunity to review your work. I have only minor comments:

- Title: Please revise to imply the main findings of your study. A statement title is more acceptable than questionable one.

Answer: Dear reviewer, we have changed the title according to your recommendation.

- Results: Figures of treatment might not be necessary.

Answer: Dear reviewer, we thank you for pointing this out and we are sure that experts in this field such as orthopaedic surgeons will immediately know what it is all about even without figures. Since the article is intended to appeal to a broad readership including general surgeons and the case presented also nicely illustrates the CWI as such, we would be happy, with your consent, not to do without the illustrations.

- Discussion: What are the implication of your findings in more general surgeon, since the readers might not be orthopaedic surgeon.

Answer: Dear reviewer, thank you for this important note. In this context, we would like to emphasize again how important the lung function is for the overall outcome. A corresponding sentence was added in the discussion.